# Modelling cell shape in 3D structured environments: A quantitative comparison with experiments

**Rabea Link**[1,2], **Mona Jaggy**[3], **Martin Bastmeyer**[3,4], **Ulrich S. Schwarz**[1,2]*

**1** Institute for Theoretical Physics, Heidelberg University, Heidelberg, Germany, **2** BioQuant, Heidelberg University, Heidelberg, Germany, **3** Zoological Institute, Karlsruhe Institute of Technology (KIT), Karlsruhe, Germany, **4** Institute for Biological and Chemical Systems, Biological Information Processing (IBCS-BIP), Karlsruhe Institute of Technology (KIT), Karlsruhe, Germany

* schwarz@thphys.uni-heidelberg.de

## Abstract

Cell shape plays a fundamental role in many biological processes, including adhesion, migration, division and development, but it is not clear which shape model best predicts three-dimensional cell shape in structured environments. Here, we compare different modelling approaches with experimental data. The shapes of single mesenchymal cells cultured in custom-made 3D scaffolds were compared by a Fourier method with surfaces that minimize area under the given adhesion and volume constraints. For the minimized surface model, we found marked differences to the experimentally observed cell shapes, which necessitated the use of more advanced shape models. We used different variants of the cellular Potts model, which effectively includes both surface and bulk contributions. The simulations revealed that the Hamiltonian with linear area energy outperformed the elastic area constraint in accurately modelling the 3D shapes of cells in structured environments. Explicit modelling the nucleus did not improve the accuracy of the simulated cell shapes. Overall, our work identifies effective methods for accurately modelling cellular shapes in complex environments.

**Data Availability Statement:** Computer code and experimental data has been deposited on zenodo (https://doi.org/10.5281/zenodo.10673295). This includes the CompuCell3D simulation scripts, the custom-written CompuCell3D linear surface plugin

## Author summary

Cell shape and forces have emerged as important determinants of cell function and thus their prediction is essential to describe and control the behaviour of cells in complex environments. While there exist well-established models for the two-dimensional shape of cells on flat substrates, it is less clear how cell shape should be modeled in three dimensions. Different from the philosophy of the vertex model often used for epithelial sheets, we find that models based only on cortical tension as a constant geometrical surface tension are not sufficient to describe the shape of single cells in 3D. Therefore, we employ different variants of the cellular Potts model, where either a target area is prescribed by an elastic constraint or the area energy is described with a linear surface tension. By

(needed to evaluate the surface energy of the cell), the python script to generate input (FE) files for SurfaceEvolver (from STL-files, which in turn can be generated from WRL-files, e.g. with MeshLab) and the python script to calculate the frequency spectrum and Delta30. On the data side, we provide all WRL-files generated with Imaris from the Zeiss image data files.

**Funding:** This work is supported by the Deutsche Forschungsgemeinschaft (DFG, German Research Foundation) under Germany's Excellence Strategy EXC 2082/1-390761711 (cluster of excellence 3DMM2O) to MB and USS. The funders had no role in study design, data collection and analysis, decision to publish, or preparation of the manuscript.

**Competing interests:** The authors have declared that no competing interests exist.

comparing the simulated shapes to experimental images of cells in 3D scaffolds, we can identify parameters that accurately model 3D cell shape.

## Introduction

The shape of animal cells is the result of active and passive intracellular and extracellular forces arising from actin polymerization, actomyosin contraction, cell adhesion, and material properties of the cell and its mechanical environment [1]. The cell membrane, a lipid bilayer with fluid-like properties, acts as the physical boundary of the cell and determines cell area and volume, but does not contribute directly to the mechanics of adherent cells. In most cell types, the actin cortex, a thin layer of actin filaments and myosin motor proteins located directly beneath the membrane, is the main mechanically relevant component [2]. The cortex actively contracts the cell surface, introducing tension gradients that enable the cell to change shape [3]. Other internal structures relevant to cell shape include the nucleus, which is anchored in the cytoskeleton, actin stress fibers, which are actomyosin filament bundles that form dynamically in response to the mechanical environment, microtubules, stiff hollow structures that provide intracellular coordination and stability, and intermediate filaments, which contribute to cell integrity and resilience against external stress, mainly in epithelial cells [4–6]. Although our understanding of each system is increasing, the interplay between the three filament systems is crucial for cell mechanics and their resulting cell shape is difficult to predict due to the complexity of the system [7].

Because cell shape is influenced by so many different factors, in principle it can be highly variable. As different cell types ensure specific functions within the organism, their shapes are optimized for these functions depending on cell type. Furthermore, cell shapes change during development and morphogenesis. While all animal cells have the ability to change their shape in principle, some cell types like epithelial cells, muscle cells or neurons show little shape changes in the somatic stage. Other cell types, mainly those which function alone in the context of connective tissue, such as keratocytes and fibroblasts, change their shape more frequently [8]. Additionally, cell shape changes are necessary during division, even for epithelial cells, that usually are tightly integrated with their neighboring cells, but round up for divison [9]. With recent technological advancements, it is now easier to obtain 3D images of cells, enabling the investigation of not only their 2D projections, but also their full 3D shape [10–12]. Concomitant with these advances in imaging, also the corresponding image processing algorithms, which formerly have been developed mainly for 2D [13, 14], now quickly improve to quantitatively evaluate cell shape also in 3D [15]. These developments now make it possible to establish a closer link to modelling and simulation of cell shape.

Mathematical modelling plays an important role in improving and validating our understanding of cell shape, especially if combined with quantitative experiments. Simulating cell shape in structured environments tests and increases our understanding of the underlying mechanisms governing cell behavior [16]. Analytical 2D cell shape models have been developed to predict cell shape on micropatterned environments using line and surface tensions [17, 18], and the notion that the contractile actin cortex is responsible for 3D cell shape is wide-spread [19, 20]. In fact this philosophy underlies the popular vertex models for epithelial sheets and 3D cell assemblies, which reduces cell mechanics to the contractility of the cellular interfaces [21, 22]. 3D cell shape simulations have relied also on more phenomenological approaches, including neural networks [23] and learned probability distributions [24], but for a mechanistic understanding, it is desirable to start from physical models of cell surface

mechanics. A popular and very versatile framework for doing so is the cellular Potts model (CPM), which combines several mechanical contributions into one energy function [25].

In this study, we employ energy-based descriptions to model cell shape in well-defined environments and compare different surface energy descriptions. In particular, we compare an explicit surface minimization to the CPM-approach, which represents different physical mechanisms in one framework. Our approach provides a robust framework for modelling complex 3D cell shapes and has the potential to improve our understanding of the fundamental principles that govern cell behavior in structured environments. Although cell shape in biological tissue is more variable than shown here, our approach identifies the elements required to address these more complex situations.

In order to quantify cellular shapes in structured environments, we utilized 3D structures manufactured with laser nanoprinting, which allowed us to create precisely controlled conditions for cell adhesion [26]. First, we compared the experimentally observed shapes of single mesenchymal NIH/3T3 cells in structured environments with minimized surfaces under volume constraint using a Fourier decomposition. We found that these shapes can differ significantly, confirming that cell shape is more complex than surfaces under constant tension (surfaces of constant mean curvature). To address this issue, we employed two additional modelling approaches, namely the CPM with elastic or linear area energies. Our simulations showed that the Hamiltonian with linear area energy outperformed the elastic area constraint in accurately modelling the shapes of cells in structured environments. We also found that explicitly modelling the nucleus did not generally improve the accuracy of the simulated cell shapes. Overall, our study provides insights into effective methods for modelling cellular shapes in complex environments.

## Materials and methods

### 3D structured environments

We manufactured structured environments for cells using 3D laser nanoprinting [26]. In this technique, a photopolymerizable resist forms radicals in the focal volume of a femtosecond-pulsed laser. The resist only polymerizes after a two-photon absorption, and because the probability for a two-photon polymerization is only high enough in the focal volume of the laser, complex structures can be printed with submicrometer precision.

The fabricated structures consisted of L-shaped, V-shaped, right-triangular and equilateral-triangle shaped patterns, see Fig 1a-a''' (design scheme) and b-b''' (electron micrographs), with 15μm high anti-adhesive columns connected by biofunctionalized cross struts of 5μm width, providing a suitable platform for cell adhesion in 3D. These geometries were selected to resemble the shapes of commonly used 2D micropatterns, which are well investigated in terms of 2D cell shapes [27, 28], while eliminating the apico-basal polarity observed in cells on substrates. Using 3D shapes with planar geometry also has the advantage of easy cell seeding and good imaging conditions.

To functionalize the structures, they were first rinsed with 70% ethanol (Carl Roth) and then dried for 30 min under UV light. Thereafter, the structures were overcoated with 200 $\mu$/ml poly-L-lysine (Sigma-Aldrich) dissolved in phosphate-buffered saline (PBS, Biochrom AG) for 1 h at room temperature and then washed three times with PBS. This was followed by incubation with 10 μg/ml fibronectin in PBS for 1 h at room temperature. The functionalized structures can be seen in Fig 1c-c'''. After washing again three times with PBS, the structures were either used directly or stored in PBS for a maximum of two days at 4°C.

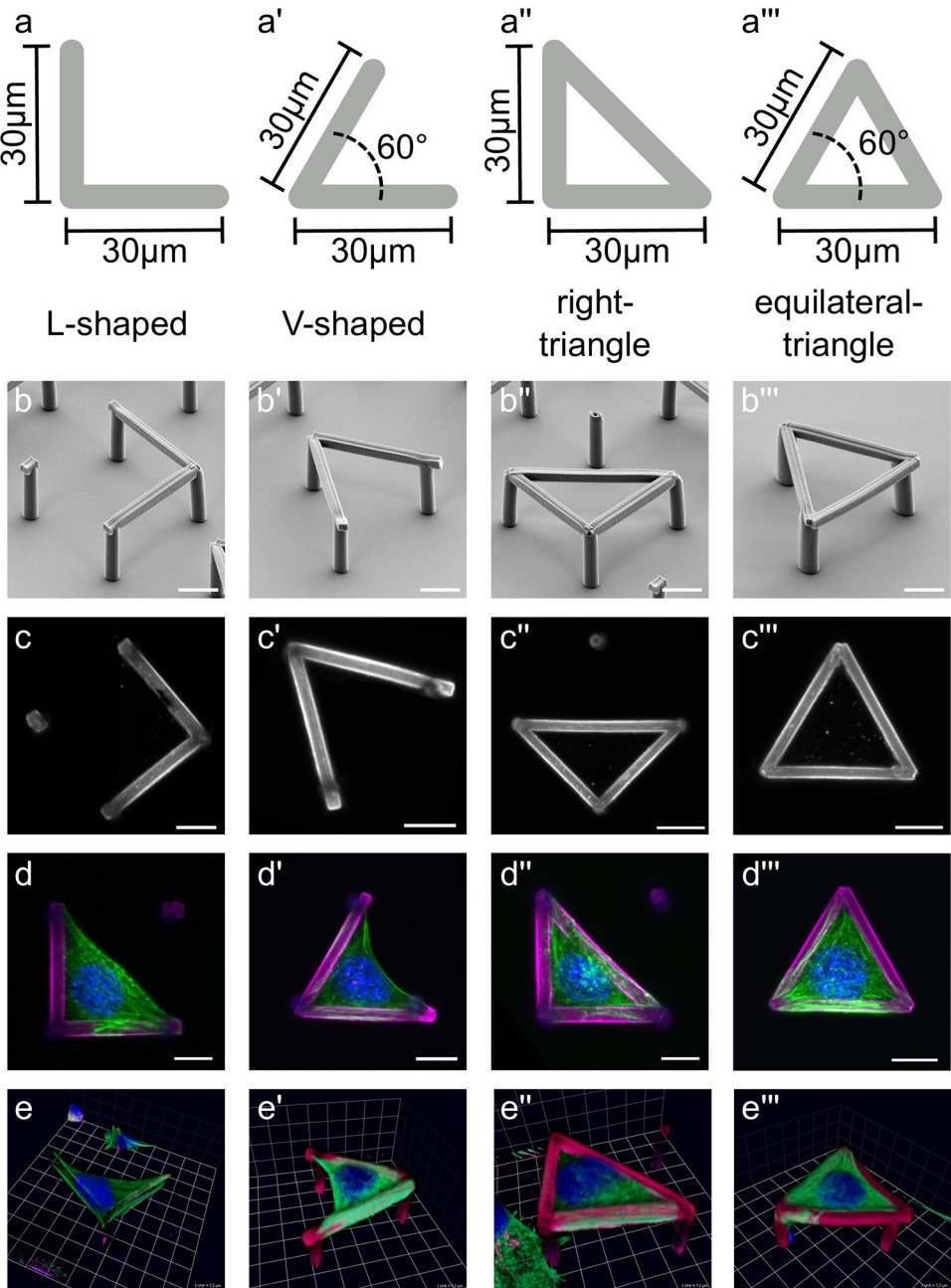

**Fig 1. Overview of the 3D printed scaffolds used in this work. a-a′′′**) Top view with dimensions of the 3D structures. Shown in gray is the contact area ("cross struts") for the cells in the L-shaped (**a**), V-shaped (**a′**), right triangle-shaped (**a′′**) and equilateral triangle-shaped (**a′′′**) structures. The width of the struts is 5 μm. The supporting columns (not depicted) have a height of 15μm. The angles not explicitly indicated are 90˚ and 45˚. **b-b′′′**) Scanning electron microscope images of the 3D structures. **c-c′′′**) Fluorescence imaging of the structures after fibronectin coating and immunohistochemical staining. **d-d′′′**) Fluoresence imaging of NIH/3T3 cells in fibronectin-coated structures after immunohistochemical staining (fibronectin = magenta, self-fluorescence of columns = blue, DAPI = blue, actin = green). **e-e′′′**) 3D reconstruction of the cells in 3D structures. The cells adhere to the cross struts, using the entire strut as the adhesion surface. Scaffold not depicted in **e**. Scale bar: 10μm.

## Cell culture

NIH/3T3 embryonic mouse fibroblast cells were cultured at standard conditions (saturated humidity, 37˚C, 5% $CO_2$) in serum containing medium and passaged three times per week to avoid contact inhibition. During passaging, cells were first rinsed twice with pre-warmed PBS and then incubated with 250 μL, 5% trypsin / 10 mM EDTA (Invitrogen) at 37˚C for 3 min to allow cells to detach from the substrate. The cell suspension was taken up in 5 mL of pre-warmed DMEM (Invitrogen) containing 10% fetal calf serum (FCS, PAA Laboratories). The serum in the medium provides saturation of trypsin. The cells were then centrifuged at 1000 rpm for 5 min. The supernatant was removed and the cell pellet resuspended in 5–10 mL of medium. Depending on the desired dilution, a certain volume was transferred to new cell culture flasks with pre-prepared tempered medium. The usual division ratios for NIH/3T3 cells is between 1:10 and 1:20.

After structure functionalization with fibronectin, NIH/3T3 mouse embryonal fibroblasts were seeded on the structures using a micromanipulator (aureka, aura optik GmbH) with an attached hydraulic manual microinjector (CellTram, Eppendorf). Cells were added to 4˚C $CO_2$ buffered F12- imaging medium (0.76 g F12 nutrient mixture (Invitrogen), 50 mL water, 25mM HEPES (Carl Roth), 1% Pen/Strep (Sigma-Aldrich), 200mM L-glutamine (Life Technologies), 10% FCS). A portion of this cell suspension was pipetted over the glass plate containing the structures, which was also coated with F12 imaging medium (4˚C) and clamped in a magnetic holder. The unheated medium reduced premature adhesion of the cells to the substrate bottom and to the glass capillary of the microinjector. The individual cells were then aspirated to a glass capillary via the microinjector by creating a negative pressure. The capillary was then conveyed over the desired structure using the micromanipulator and the cell was transferred. After all structures were occupied, the temperature was raised to 37˚C to accelerate adhesion of the cells to the structures.

Protein staining was performed immunologically on fixed samples in a humidity chamber. Cells were fixed at room temperature for 10 min with 37˚C tempered 4% PFA (Sigma-Aldrich) in PBS. This was followed by permeabilization of the cell membrane by washing three times with 0.1% TritonX-100 (Carl Roth) in PBS for 5 min, followed by incubation with anti-fibronectin (BD Transduction Laboratories, 1:500) for 1 h at room temperature or at 4˚C. All antibodies and staining substances were diluted in 1% BSA (Bovine Serum Albumin) in PBS. This was followed by three wash steps for 5 min each with PBS. The fluorescent secondary antibodies anti mouse AF 647 (Life Technologies, 1:400) as well as DAPI (Roth, 1:2000) and fluorescently coupled phalloidin AF 488 (Life Technologies, 1:200) were then applied for 1 h at room temperature. After another wash step with PBS, the samples were embedded in 1% n-propyl gallate (Sigma-Aldrich) in Mowiol (Hoechst) and stored at 4˚C.

3D images of the cells were taken on the LSM 510 Meta (Zeiss) and the Axio Imager.Z1 with ApoTome (Zeiss) at 37˚C, compare Fig 1d-d‴. The 3D shapes were extracted from the actin, DAPI and fibronectin staining as triangulated meshes (WRL-files) using Imaris (Bitplane), compare Fig 1e-e‴.

## Constant Mean Curvature (CMC) surfaces

The actin cortex is a network of actin filaments and myosin motors positioned beneath the cell membrane, which contracts the cortex and influences cell shape. Assuming constant tension throughout the cortex, cell shapes can be characterized as fixed-volume objects that minimize their surface area, representing an approach that does not consider any contributions from the bulk of the cell. This approach is applicable to cells in suspension, which tend to be spherical in shape. For cells on adhesive stripes, their shapes can be described as a wetting process

governed by surface tensions [29]. However, a quantitative 3D comparison between experimentally observed cell shapes and shapes obtained from the described minimal surface model has not yet been conducted to our knowledge.

Surface minimization under volume constraints leads to surfaces with constant mean curvature (CMC) [30]. The mathematical problem of finding the minimum energy shape for a given boundary is called the Plateau problem. To assess the validity of these assumptions for cells in structured environments, we perform a comparison between experimentally observed cell surfaces and minimized surfaces with the same volume. The discrepancy between the observed and minimized cell shapes provides a measure of the degree to which factors beyond constant surface tension due to the actin cortex contribute to cell shape in structured environments.

The triangulated surfaces obtained from the experimental data were analyzed using SurfaceEvolver (Version 2.70) [31], a software that utilizes gradient descent to minimize surfaces under forces and constraints. The triangulated meshes obtained from the image processing software Imaris were simplified using quadratic edge collapse decimation [32] and then used as inputs to SurfaceEvolver. The observed volume was fixed, as well as all points attached to the 3D printed scaffold. It was assumed that the surface tension is constant everywhere. The surfaces were minimized using the gradient descent method, and the minimized triangulated meshes were exported for further analysis.

## Cellular Potts Model (CPM)

The CPM is a well established modelling framework to describe single cells or cell collectives. Introduced by Glazier, Graner and Hogeweg in the 1990s to model differential adhesion [33, 34], CPMs are predominantly used to describe phenomena arising due to cellular interaction [35–37], but they have also been used to describe single dynamics on micropatterns [18, 38–40]. In contrast to the explicit surface models and similar to phase field models [41], they include also contributions from bulk elasticity. In contrast to hydrodynamic or active gel models, however, their focus is on cell surface mechanics [25].

In detail, cells in the CPM are modeled on a lattice where each lattice site can be occupied by a generalized cell $\sigma$ of a predefined cell type $\tau$. In our case, possible types are cytoplasm, nucleus, medium as well as adhesive and non-adhesive scaffold; different from e.g. CPM-simulations on cell sorting, we do not have multiple biological cell types. A generalized cell typically occupies many lattice sites. The Hamiltonian $\mathcal{H}$ is an energy functional that defines the energy for all possible lattice configurations. During the simulation, a modified Metropolis algorithm is used to minimize the total energy of the system. The algorithm attempts to update the configuration of the system by selecting a lattice site and trying to change its state to that of a neighboring generalized cell. The change is accepted with a probability given by the Metropolis rule, which depends on the energy difference between the new and the old configurations. By repeating this process many times, the system evolves to a state that minimizes energy, but is also able to cross local energy barriers [42]. Given the relatively simple shapes considered here, our simulations always find a unique steady state, thus we did not make use of simulated annealing procedures [39].

Choosing the appropriate Hamiltonian to describe biological systems has been a longstanding question in the field. Early formulations of the Hamiltonian included an elastic volume constraint and interaction energies. In order to more accurately model cell behavior, an elastic surface constraint was added. Additionally, the nucleus can be modeled explicitly as a compartmentalized cell with an elastic constraint to ensure that the nucleus is close to the cell

center of mass:

$$\mathcal{H} = \sum_{\text{Cytoplasm, Nucleus}} \lambda_{V_\tau} \cdot \left( V(\sigma) - V_{T_\tau} \right)^2 + \sum_{\text{Cell, Nucleus}} \lambda_{A_\tau} \left( A(\sigma) - A_{T_\tau} \right)^2$$

$$+ \sum_{<\mathbf{x},\mathbf{x}'>_N} J_{\tau(\sigma(\mathbf{x})),\tau(\sigma(\mathbf{x}'))} \left( 1 - \delta(\sigma(\mathbf{x}), \sigma(\mathbf{x}')) \right) \tag{1}$$

$$+ \lambda_N \left( \mathbf{x}_{\text{Cell}} - \mathbf{x}_{\text{Nucleus}} \right)^2.$$

Here the first term contains the elastic volume energies for the generalized cell types cytoplasm and nucleus. The second term describes the elastic surface energies of the cell and the nucleus; note that the cell surface is not identical with the cytoplasm surface, which would contain also the interface to the nucleus. The scaffold is implemented as a fixed generalized cell type. The inclusion of an elastic volume constraint in the Hamiltonian is motivated by the constant volumes of cell compartments in biological systems, and the need to model dynamics with some flexibility to avoid "lattice freezing" while ensuring that the simulated cells do not disappear, which would reduce the energy of the system but does not align with biological reality. Conversely, the elastic area constraint is not strictly required for the simulation, but its inclusion provides an additional constraint that allows more flexibility in the parameter selection of the interaction energies $J$ [42]. The third term in the Hamiltonian represents the interaction energy localized at the cell interfaces. It is computed by summing over all voxels $\mathbf{x}$, $\mathbf{x}'$ within the neighborhood $N$, which for $N = 1$ are the 6 directly adjacent voxels, for $N = 2$ it includes the 8 diagonally adjacent voxels and so on. If the voxels $\mathbf{x}$, $\mathbf{x}'$ belong to different generalized cells $\sigma$, their interaction energy $J$ is added to the total energy of the system. Summing over the neighborhood diminishes the effect of the anisotropic lattice and in addition introduces a bulk element into the model. The value of $J$ can be positive or negative and depends on the cell types of the neighboring voxels. To ensure that the nucleus remains inside the cell, an elastic constraint with strength $\lambda_N$ is added between the cell center of mass $\mathbf{x}_{\text{Cell}}$ and the center of mass of the nucleus $\mathbf{x}_{\text{Nucleus}}$.

Instead of the elastic area constraint that follows from the assumption that the cell surface is regulated to a specific value throughout the experiment, one can describe the surface with a linear area term, which follows from the assumption that increasing the cell surface is always possible, albeit for a certain energy cost [36, 39]:

$$\mathcal{H} = \sum_{\text{Cytoplasm, Nucleus}} \lambda_{V_\tau} \left( V(\sigma) - V_{T_\tau} \right)^2 + \lambda_{A_{\text{Cell}}} A(\text{Cell}) + \lambda_{A_{\text{Nucleus}}} \left( A(\text{Nucleus}) - A_{T_{\text{Nucleus}}} \right)^2$$

$$+ \sum_{<\mathbf{x},\mathbf{x}'>_N} J_{\tau(\sigma(\mathbf{x})),\tau(\sigma(\mathbf{x}'))} \left( 1 - \delta(\sigma(\mathbf{x}), \sigma(\mathbf{x}')) \right) \tag{2}$$

$$+ \lambda_N \left( \mathbf{x}_{\text{Cell}} - \mathbf{x}_{\text{Nucleus}} \right)^2.$$

Here, the volume, interaction and nucleus surface and centering energy terms are similar to Eq 1, but the elastic area energy term of the cell is replaced with a linear area energy, where the strength is determined by $\lambda_{A_{\text{Cell}}}$. Note that here, $\lambda_{A_{\text{Cell}}}$ has different dimensions compared to Eq 1 because it represents different physics.

The simulations were implemented using CompuCell3D (version 4.1.1) [43]. Cell shape was initialized as a triangular prism pyramid positioned between the adhesive sites and it was checked that the final shape did not depend on the details of the initial conditions. The static cell shapes used for further analysis were obtained by averaging over the last 500 Monte Carlo steps of the 2000 Monte Carlo steps simulation.

### 3D spherical harmonics analysis

To analyze and quantitatively compare 3D cell shapes obtained from experiments and simulations, we employed a 3D spherical harmonics analysis. Spherical harmonics form a complete set of orthonormal functions and can be used as an orthonormal basis for describing 3D shapes. This approach enables precise, translation-invariant, scale-invariant, and rotation-invariant description of 3D shapes and has been previously used to analyze biological shapes [44–48].

Following the approach in [47], we first convert the boundary shapes for simulated and experimentally observed cell shapes from Cartesian coordinates to spherical coordinates, exploiting the fact that all our shapes are star-shaped and thus can be mapped bijectively to the unit sphere. We then sample the data on a regular grid and use the Driscoll and Healy sampling theorem [49] implemented in pyshtools (version 4.10) [50] to calculate the spherical harmonics:

$$Y_l^m(\theta, \phi) = k_{l,m} P_l^m(\cos\theta) e^{im\phi}, \tag{3}$$

where $l$ and $m$ are the degree and the order, respectively. $k_{l,m}$ is the normalization and $P_l^m$ are the associated Legendre polynomials. The spherical harmonics as defined in Eq 3 define an orthonormal basis, thus any scalar function $f(\theta, \phi)$ on a sphere can be expressed as a sum of the spherical harmonics:

$$f(\theta, \phi) = \sum_{l=0}^{\infty} \sum_{m=-l}^{l} \hat{f}(l, m) Y_l^m(\theta, \phi). \tag{4}$$

Here, $\hat{f}(l, m)$ are the harmonic coefficients given by

$$\hat{f}(l, m) = k_{l,m} \int_0^{\pi} \int_0^{2\pi} e^{-im\phi} f(\theta, \phi) P_l^m(\cos\theta) \sin\theta \, d\phi \, d\theta. \tag{5}$$

We normalize the harmonic coefficients $\hat{f}(l, m)$ with respect to the first-order ellipsoid $\hat{f}(0, 0)$ and use the normalized coefficients $\hat{f}_n(l, m)$ to calculate the rotation-invariant frequency spectrum $F(l)$ as a quantitative shape measure

$$F(l) = \sum_{m=-l}^{l} \hat{f}_n^2(l, m) \tag{6}$$

Now, we can calculate a measure for the difference $\Delta_{l_{\max}}$ between two shapes $a$ and $b$ from the corresponding frequency spectra:

$$\Delta_{l_{\max}}(a, b) = \sqrt{\sum_{l=0}^{l_{\max}} \left( F_a(l) - F_b(l) \right)^2}. \tag{7}$$

In the subsequent analysis, we will use the first 30 degrees of the respective frequency spectra to compare shapes ($\Delta_{30}$), because higher modes are not resolved in our data and do not contribute significantly.

## Results

### Characterization of experimentally observed shapes

We first analyzed the shapes of $n = 6$ cells in L-shaped structures spanning between two fibronectin coated cross struts. A representative cell can be seen in Fig 1d and 1e and S1 Video.

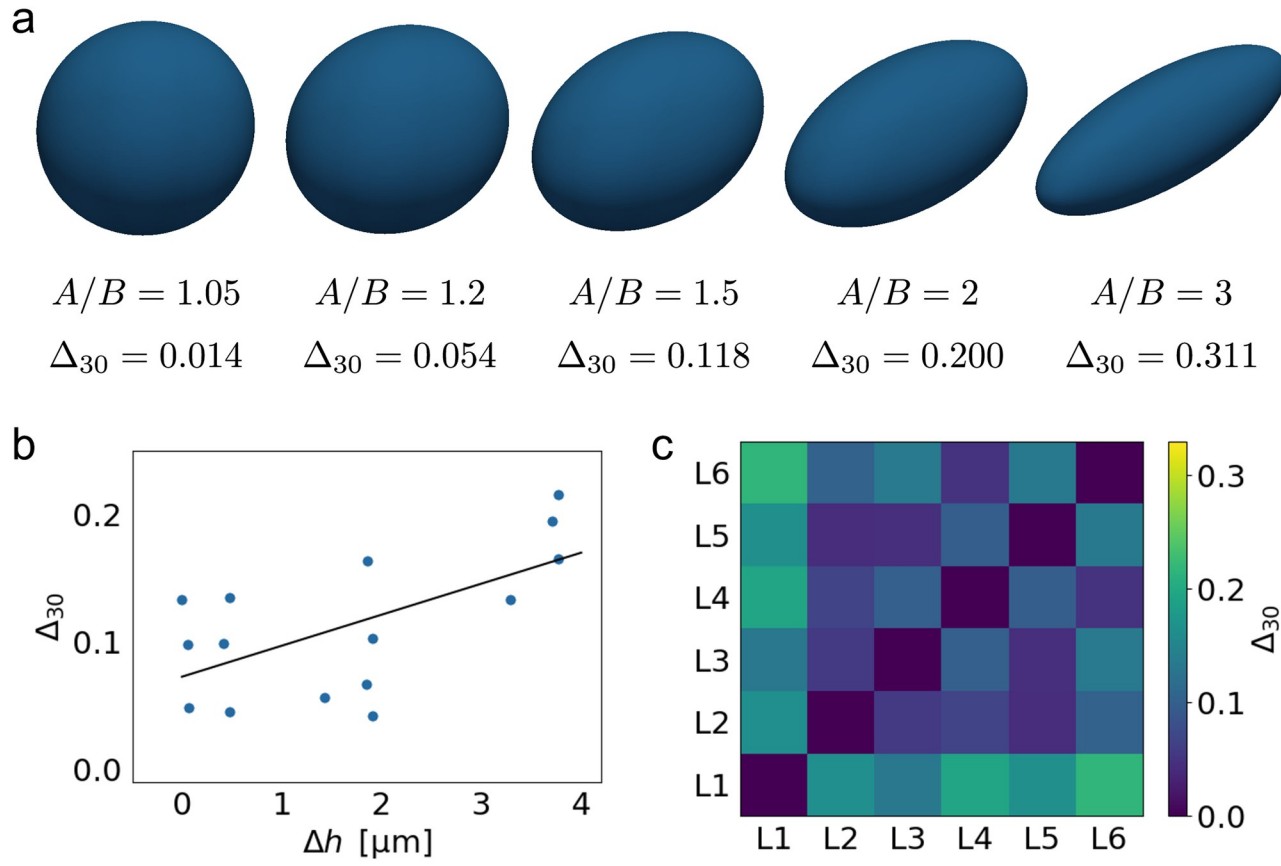

**Fig 2. Comparison of cell shapes on L-scaffolds. a)** The shape difference measure $\Delta_{30}$ between a sphere and ellipsoids with the same volume, but different ratios between the semi-axes $A$ and $B$ is shown as reference. **b)** Difference in height of the experimentally observed cell shapes $\Delta h$ versus the difference between the cell shapes $\Delta_{30}$. The fitted line indicates a weak correlation ($R = 0.62$). **c)** Heatmap of the difference $\Delta_{30}$ between the experimentally observed cell shapes within the L-class.

Visual inspection of the extracted cell shapes revealed remarkable similarity across all cells. Notably, the cells bridged along the free area between the beams, forming invaginated arcs—a well-established phenomenon observed for cells on both 2D microstructured surfaces and 3D structures [27, 51]. In some cases, we observe w-shaped invaginations due to the nucleus. Typical values for the shape difference measure $\Delta_{30}$ within this class were around 0.1. To get an impression of the meaning of this value, in Fig 2a we record the values for the differences between a sphere and ellipsoids with the same volume, but different aspect ratios $A/B$. We see that the typical value of 0.1 corresponds to an aspect ratio below 1.5. We also observed a correlation between $\Delta_{30}$ and the difference in height of the experimentally observed cell shapes $\Delta h$, compare Fig 2b. For a systematic overview, we therefore ordered the cells according to height and compute all $\Delta_{30}$-values, compare Fig 2c. This analysis demonstrates how the shape difference $\Delta_{30}$ varies within one class (the L-shapes) and establishes a reference for the comparison between experimentally observed and simulated shapes.

Similarly, the shapes of $n = 7$ cells in V-shaped structures were analyzed, see Fig 1d′ and 1e′ and S2 Video, where we again observe invaginated arcs. In this case, the nucleus does not interfere with the formation of the arc, and we do not observe w-shaped arcs. Additionally, we analyzed the shapes of $n = 3$ cells in right triangle structures, see Fig 1d″ and 1e″ and S3 Video, and of $n = 4$ cells in equilateral triangle structures, see Fig 1d‴ and 1e‴ and S4 Video. In these

structures, the cells also adhere to the additional cross strut as compared to the L-shaped structure or the V-shaped structure respectively. In these cases, we do not find invaginated arcs as the cells fill the volume between the struts.

## Comparison to CMC-surfaces

We employed Imaris to extract triangulated meshes of the experimentally observed shapes (scaffold, cytoplasm and nucleus) and then united cytoplasm and nucleus to obtain one compact object whose surface can be minimized, see the blue surfaces in Fig 3a-a'''. To this end we used the so-obtained triangulated mesh and fixed vertices close to the structures. Then, we used SurfaceEvolver to minimize the shapes under a constant volume constraint. Upon visual inspection, we found significant differences between the experimentally observed cell shapes and the resulting minimal energy shapes for the L- and V-shaped structures, compare Fig 3a and 3b and Fig 3a' and 3b', indicating that minimizing area under a volume constraint is not sufficient for accurately describing cell shape in these cases. The observed cells in the scaffolds stretch between adhesive areas and maintain a roughly constant thickness, while the minimal energy surfaces form more sphere-like shapes with two thinly stretched extensions attached to the scaffold due to the imposed boundary conditions. This finding is not surprising since spheres have the lowest surface-to-volume ratio, thus the surfaces are becoming locally sphere-like to minimize their surface area. In general, these shapes seem to have characteristics of unduloids, which are a class of axisymmetric CMC-surfaces. On the other hand, for cells in triangular shaped structures adhering to all three cross struts, the difference between the experimentally obtained and minimized surfaces are smaller, compare Fig 3a'', 3b'' and 3a''' and 3b'''. The surfaces are smoother, but cell-scale surface changes are not found.

Landmark values of the surfaces before and after minimization quantify the shape differences. We compare the surface area reduction $\Delta A$ and the changes in reduced volume $\Delta v$. The reduced volume $v$, which quantifies how much a shape differs from a perfect sphere, was calculated from $v = V/(4\pi R_0^3/3)$, where $V$ is the volume of the cell and $R_0$ is computed from the surface area $A$ of the cell as $A = 4\pi R_0^2$. A reduced volume of 1 indicates a perfect sphere, while all other shapes have a lower reduced volume. The differences in area and reduced volume are illustrated in Fig 3c-c'''. For cells in the L-shaped structures, the surface area was reduced by (10±3)% during the minimization process on average, for the V-shaped structures the reduction was (9±3) %. Much smaller reductions were found for cells in triangular structures, cell area in the right angle triangle was reduced on average by (5±2) % and in equilateral triangles the difference is (2±1) %. As expected, surface area always went down under minimization, even if this is hard as for the closed triangular frames, but the large values obtained for the L- and V-frames indicate that 3D cell shape is not determined only by minimization of surface tension.

For the reduced volume $v$, we find similar differences. After minimization, the reduced volume of cells in L-shaped structures increased by an average of (14±3) % when compared to the reduced volume before minimization. For cells in V-shaped structures, this difference was (11±4) %, whereas for cells in right angle triangles, the average difference of $\Delta v$ was found to be (6±2) % and for cells in equilateral triangles (4±1) %. In general, an increase was to be expected because the surfaces develop more undulations during minimization.

These significant variations for cells that form invaginated arcs underscore the notion that 3D cell shape in structured environments is more complex than a surface that has been minimized under a volume constraint due to an isotropically contracting actin cortex.

To quantify these differences further, we subsequently computed the spherical harmonics coefficients of the obtained experimental and minimized shapes. Because the spectrum quickly

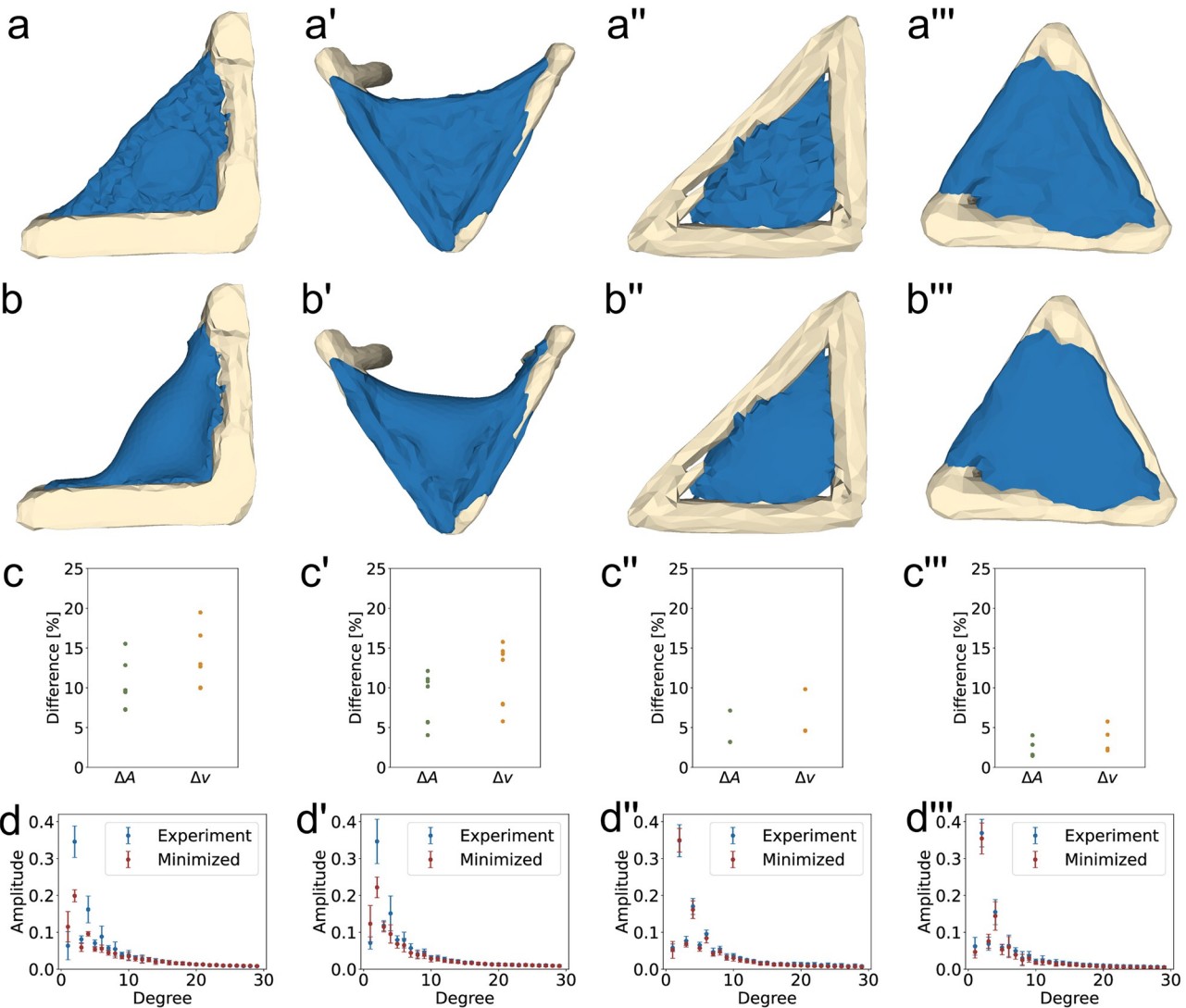

**Fig 3. Comparison between experimentally observed cell shapes and their minimized surfaces. a-a′′′)** The surfaces of the cells in structured environments. **b-b′′′)** Minimized surfaces in the 3D structures. Mesh points attached to the structure were fixed during minimization. **c-c′′′)** Area ($\Delta A$) and reduced volume ($\Delta v$) differences between the experimentally observed and minimized surfaces. **d-d′′′)** Frequency spectrum of the experimentally observed (blue) and minimized (red) surfaces. **d)** $\Delta_{30} = 0.175$. **d′)** $\Delta_{30} = 0.148$. **d′′)** $\Delta_{30} = 0.026$. **d′′′)** $\Delta_{30} = 0.031$. Movies of the 3D reconstructions and the minimized shapes can be found in S1–S4 Videos.

decays and does not represent very fine details, as expected due to the finite resolution of the imaging data, we only consider modes up to $l_{max} = 30$. The frequency spectrum of the spherical harmonics for experimentally observed and minimized shapes can be seen in Fig 3d-d′′′. The dipole moment of the spherical harmonics, represented by the amplitude of the first degree in the frequency spectrum, exhibits a low value. This result is anticipated, given the observed cell shape's lack of symmetry with respect to a single axis. On the other hand, the second degree of the frequency spectrum displays the highest amplitude across all observed cell shapes, corresponding to the quadrupole moment that characterizes a distribution with two perpendicular axes of symmetry. Additionally, the fourth spherical harmonic also has a pronounced peak in the frequency spectrum, representing a six-axis symmetric distribution.

With the spherical harmonics analysis of the minimized structures, we are able to quantify the difference between experimentally observed shapes and what we would expect from minimal surfaces. The difference between the average experimentally observed and minimized amplitude is $\Delta_{30} = 0.175$ for cells in L-shaped structures. For cells in V-shaped structures, we find $\Delta_{30} = 0.148$. In both cases, there is a large difference in the low degrees of the frequency spectrum. The amplitude of the first degree of the frequency spectrum for the minimized shapes is larger than the experimental amplitude, while the amplitudes of the second and forth degrees, which are prominent in the experimental shapes, are lower. This confirms the increased sphericity of the minimized shapes. When comparing the experimentally observed cell shapes to the minimized shapes in triangular structures, the difference is much smaller; for cells in the right angle triangles, we find $\Delta_{30} = 0.026$, and for cells in equilateral triangles we find $\Delta_{30} = 0.031$. Both values are much lower than for structures with invaginated arcs, highlighting that the accuracy of cell shape prediction using constant mean curvature depends on the structured environment.

With this we show that 3D cell shape in structured environments is not always adequately described by minimized surfaces. We conclude that a constant surface tension due to a contractile actin cortex is not enough to describe 3D cell shape, especially for cells in 3D with invaginated arcs.

## Comparison to CPM-shapes

The CPM can be used to simulate the behavior of cells in complex geometries, such as the scaffolds used in the experiments described above. By optimizing the parameters of the model it is possible to simulate cell shapes that closely resemble those observed experimentally. Parameters influencing the cell shape are the surface energy strength $\lambda_{A_\tau}$, which represents the energy required to change the cell surface area, as well as the interaction energy between the cytoplasm and the medium $J_{\text{cytoplasm,medium}}$, which represents the energy required to expand the cell surface into the surrounding medium. The neighbor order $N$ is a parameter that determines the extent of the interactions between neighboring voxels. Choosing an appropriate neighbor order is crucial in CPM simulations, as it diminishes the effect of the underlying lattice and defines how much of the surrounding affects the cell [25]. Other parameters do not change cell shape much as long as their value is chosen within a reasonable range. The volume parameters $\lambda_{V_\tau}$ and $V_{T_\tau}$ are used to ensure that the volume of the cytoplasm and the nucleus stay close to the target volume $V_{T_\tau}$ throughout the simulation, however changing the constraint strength $\lambda_{V_\tau}$ within an appropriate range does not change the cell shape. The interaction parameters between cytoplasm and scaffold $J_{\text{cytoplasm,scaffold}}$, as well as cytoplasm and nucleus $J_{\text{cytoplasm,nucleus}}$, are chosen to ensure the cell adheres to the scaffold and the nucleus is surrounded by cytoplasm. The simulation temperature is fixed at $T = 100$ throughout the simulations, which is a standard choice in CPM-simulations and sets the energy scale. All parameters used in the CPM-simulations can be found in S1 Table. We note that while most parameters used here are standard choices, the strongly negative interaction parameter between cells and scaffolds is special to our work and represents the fact that adhesion drives the shape changes, similar to earlier CPM-simulations for 2D [18, 39].

In the following we compare the results of the CPM-simulations to the experimental results for the L-shaped scaffolds ($n = 6$), compare the representative cell shown in Fig 1d and 1e. The corresponding spectra were averaged to obtain the average spectrum of an experimental cell in a L-scaffold, compare Fig 3d. The CPM-simulation resulted in one averaged shape, the corresponding spectrum was calculated and then Eq 7 was used to calculate $\Delta_{30}$ between these shapes. In Fig 4, the measure of shape difference $\Delta_{30}$ between the experimentally observed cell

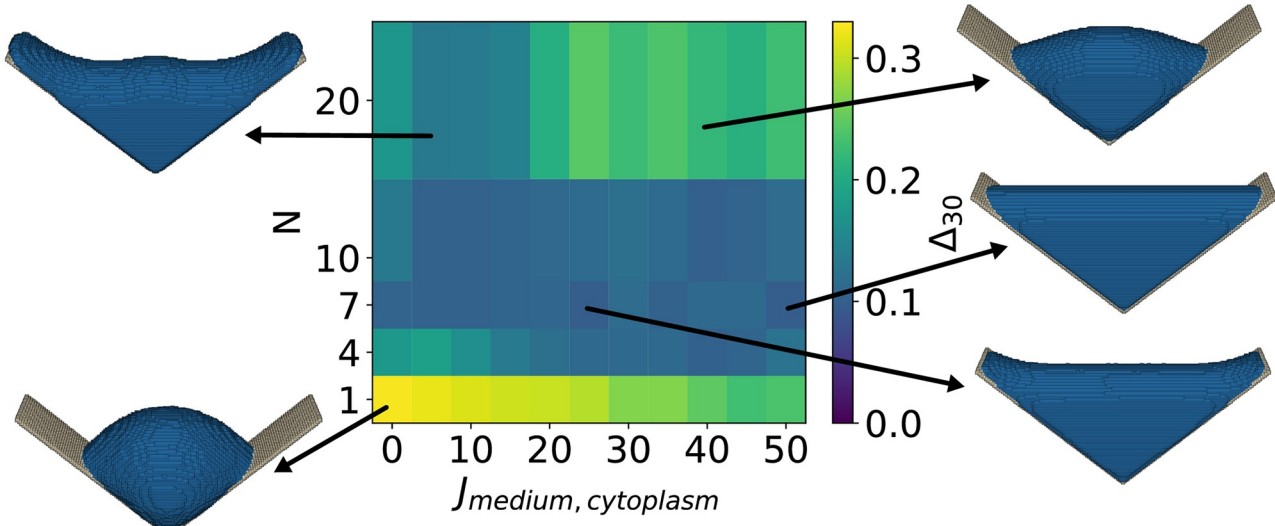

**Fig 4. Cell shape difference $\Delta_{30}$ between the experimental and simulated cell shapes as a function of neighbor order and interaction energy.** Neighbor order $N$ defines the set of neighboring lattice sites that interact with a given lattice site, and interaction energy $J_{\text{medium,cytoplasm}}$ defines the interaction strength between the cytoplasm and the surrounding medium. Simulated cell shapes for exemplary parameter choices are presented. The minimum cell shape difference $\Delta_{30} = 0.097$ is found for $N = 7$ and $J_{\text{medium,cytoplasm}} = 50$, the corresponding cell shape is depicted on the right middle.

shapes in the L-scaffolds and the cell shapes simulated with the elastic area Hamiltonian (Eq 1) is shown for varying neighbor order $N$ and interaction energy $J_{\text{medium,cytoplasm}}$. Remarkably, for a large parameter range of $J_{\text{medium,cytoplasm}}$ and intermediate neighbor order $N$, significantly lower values of $\Delta_{30}$ are obtained than for the constant mean curvature shapes, indicating that the CPM better captures the physical shape determinants of the 3D shapes.

From Fig 4, we also note that the neighbor order parameter effects the simulated cell shape more than the interaction energy $J_{\text{medium,cytoplasm}}$. When only the nearest neighbors are considered in the calculation of the interaction energy ($N = 1$), then the cell partially detaches from the scaffold due to a reduced energy gain from cytoplasm-scaffold interactions. Additionally, both the cytoplasm and nucleus become more spherical. For intermediate neighbor orders, $4 \leq N \leq 10$, the simulated cell shapes resemble the experimentally observed shapes for a large parameter range of the interaction energy $J_{\text{medium,cytoplasm}}$, leading to low values of $\Delta_{30}$, with the minimum being at $\Delta_{30} = 0.097$, for $J_{\text{medium,cytoplasm}} = 50$ and $N = 7$. Increasing the neighbor order further to $N = 20$ leads to shapes that deviate more from the experimentally observed cell shapes. When the interaction energy $J_{\text{medium,cytoplasm}}$ is low, the shape becomes w-shaped and resembles that of minimized surfaces because the energy gain from cytoplasm-scaffold interactions is large due to the high neighbor order. At the same time, the remaining cytoplasm volume reduces its energy by becoming spherical, leading to a w-shape. However, when the interaction energy $J_{\text{medium,cytoplasm}}$ is sufficiently increased, the cost of a less spherical shape becomes higher than the gain from the cytoplasm-scaffold interaction, resulting in more spherical shapes and increasing the value of $\Delta_{30}$ again. The minimum of $\Delta_{30}$ is reached for $N = 7$, which is a neighbor order with a low perimeter scaling error due to the underlying lattice $J_{\text{medium,cytoplasm}}$. Therefore, we fix the neighbor order to $N = 7$ in the following simulations.

One of the long-standing questions in CPM-type simulations is the selection of the appropriate Hamiltonian. Specifically, when it comes to surface area, the two commonly used

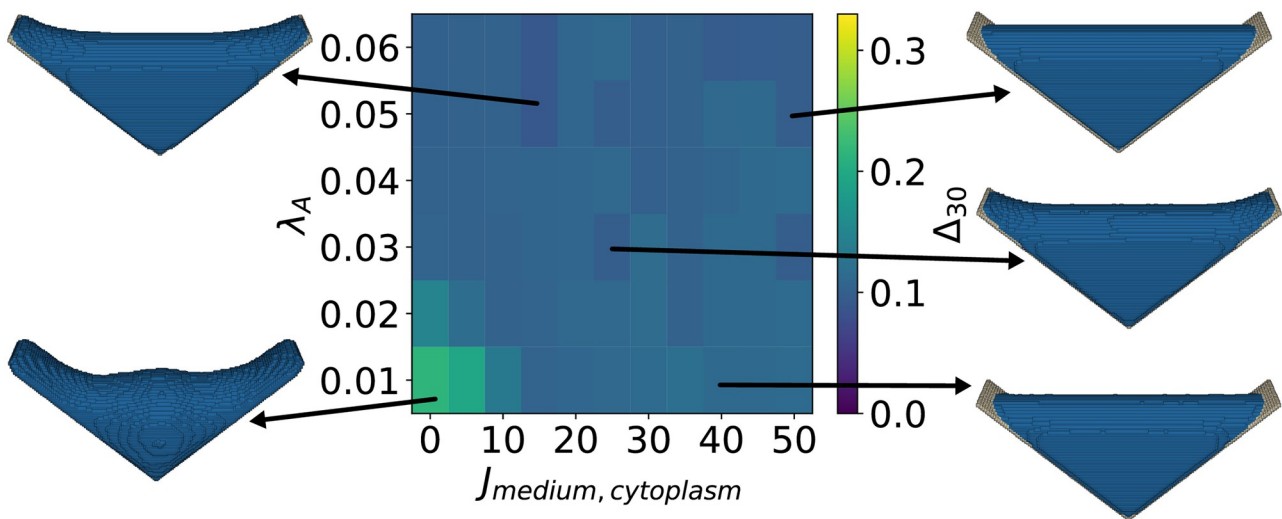

**Fig 5. Cell shape difference $\Delta_{30}$ between the experimental and simulated cell shapes obtained with the elastic area energy.** Differences are shown as a function of surface energy constraint $\lambda_A$ and interaction energy $J_{\text{medium,cytoplasm}}$. Simulated cell shapes for exemplary parameter choices are presented. The minimum cell shape difference $\Delta_{30} = 0.090$ is found for $J_{\text{medium,cytoplasm}} = 15$ and $\lambda_A = 0.05$, the corresponding simulated cell is depicted on the top left. A movie of the simulation can be seen in S5 Video.

Hamiltonians are either based on the assumption that the surface of a cell is approximately constant (possibly due to homeostatic regulation by the membrane), thus describing the area energy with an elastic constraint, see Eq 1, or that increasing surface area always costs energy for the cell (possibly because it has to work against actomyosin contractility and membrane tension), leading to a linear area energy, see Eq 2. Comparing simulated cells with experimentally observed cell shapes allows for the direct comparison between the two approaches. Cell shapes simulated with either Hamiltonian, shown in Fig 5 for elastic area energy and in Fig 6 for linear area energy, closely resemble the experimentally observed cell shapes for a wide range of parameter values, vastly outperforming the minimized surface cell shapes.

The cell shapes obtained with the elastic area Hamiltonian (see Eq 1 and Fig 5) are visually close to the experimentally observed cell shapes for a large parameter range. The minimum $\Delta_{30} = 0.090$ is found for $J_{\text{medium,cytoplasm}} = 15$ and $\lambda_A = 0.05$. Similar cell shapes are obtained for a large parameter range. A simulation with the elastic area Hamiltonian (Eq 1) can be seen in S5 Video. Increasing the interaction energy $J_{\text{medium,cytoplasm}}$ reduces the interface between cytoplasm and the medium, resulting in partially occupied scaffolds. For a low elastic area constraint $\lambda_A$ and interaction energy $J_{\text{medium,cytoplasm}} = 0$, the simulated cell shape visually differs from the experimentally observed cells and resembles again a w-shape.

The minimum found with the linear area Hamiltonian in Eq 2 is even smaller, with $\Delta_{30} = 0.039$ for $J_{\text{medium,cytoplasm}} = 0$ and $\lambda_A = 500$, see Fig 6 and S6 and S7 Videos. The simulated cell shape that resembles the experiments best is triangular shaped and without completely invaginated arcs, however the thickness to length ratio of the cell closely resembles that of the experimentally observed shapes. Surprisingly, the simulated cell shape resembles the experimentally observed shapes best when the interaction energy approaches $J_{\text{medium,cytoplasm}} = 0$. For a cell that is surrounded by medium only, increasing $J_{\text{medium,cytoplasm}}$ has a similar effect to a linear area constraint, as the surface of the cell corresponds to the interaction area between cell and medium. In the case of structured environments, the interaction energy only acts on part of the cell surface, while the whole cell surface is relevant for the area energy. For the linear area

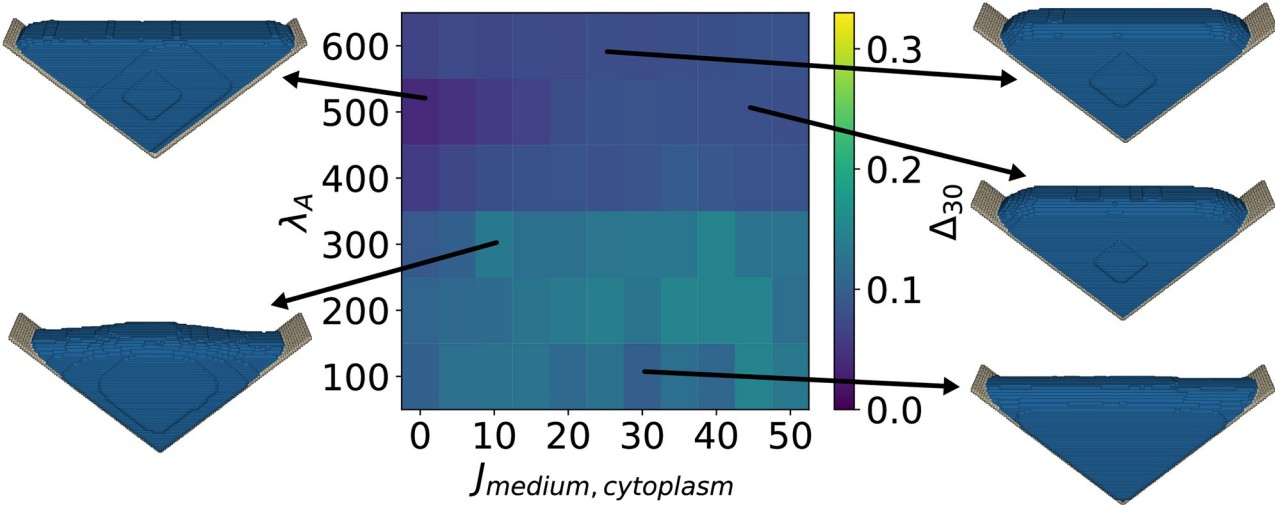

**Fig 6. Cell shape difference $\Delta_{30}$ between the experimentally observed cell shapes and simulated cell shapes obtained with the linear area energy.** Differences are shown as a function of surface energy constraint $\lambda_A$ and interaction energy $J_{medium,cytoplasm}$. Simulated cell shapes for exemplary parameter choices are presented. The minimum cell shape difference $\Delta_{30} = 0.039$ is found for $J_{medium,cytoplasm} = 0$ and $\lambda_A = 500$, the corresponding simulated cell is depicted on the top left. A movie of the simulation and final shape can be seen in S6 and S7 Videos.

Hamiltonian, the scaffold is only partially covered as increasing the cytoplasm surface area always costs energy. It is surprising that the surface corresponding best to the experiments is found for the linear area Hamiltonian simulation with interaction energy $J_{medium,cytoplasm} = 0$, as this is the simulation with the fewest parameters.

Cells with nuclei in square, V-shaped, right-angle and equilateral-triangle structures were simulated with the parameter set that performed best for the L-shaped structure (linear area Hamiltonian (Eq 2), $\lambda_A = 500$, $J_{medium,cytoplasm} = 0$), see Fig 7. For the right angle triangular structure, we find very good agreement between the experimentally observed shapes and the simulation with $\Delta_{30} = 0.053$. However, for the equilateral triangle and the V-shaped structure, which differ more from the L-shaped structure than the right angle triangle, we find $\Delta_{30} = 0.141$ and $\Delta_{30} = 0.136$, which is worse than the results for the L-shape, where we find $\Delta_{30} = 0.039$. These differences show that it is difficult to find universal parameters for the CPM that accurately predict cell shape in differently structured environments.

From the experimental data, we find that the nucleus is significantly deformed by the cell cytoskeleton, but does not have a large effect on cell shape. We include the nucleus explicitly in our CPM simulations, and do not find a visual effect of the nucleus on cell shape in the simulations. In S1 Appendix, we simulate cells without explicit nucleus representation in the CPM, and find only minor cell shape changes.

## Discussion

Due to recent technical advances in imaging, an increasing amount of experimental data is three-dimensional (3D). This adds to the need for 3D single cell models to explain and predict cell shape. Combining cell culture with structured environments opens the door for a detailed quantitative comparision between theory and experiment [16, 26]. In this work, we compared different approaches to describe cell shape in 3D. We used surface minimization and two main variants of the cellular Potts model (CPM) to investigate how much simulated shape predictions differ from experimentally observed cell shapes in precisely defined environments. We

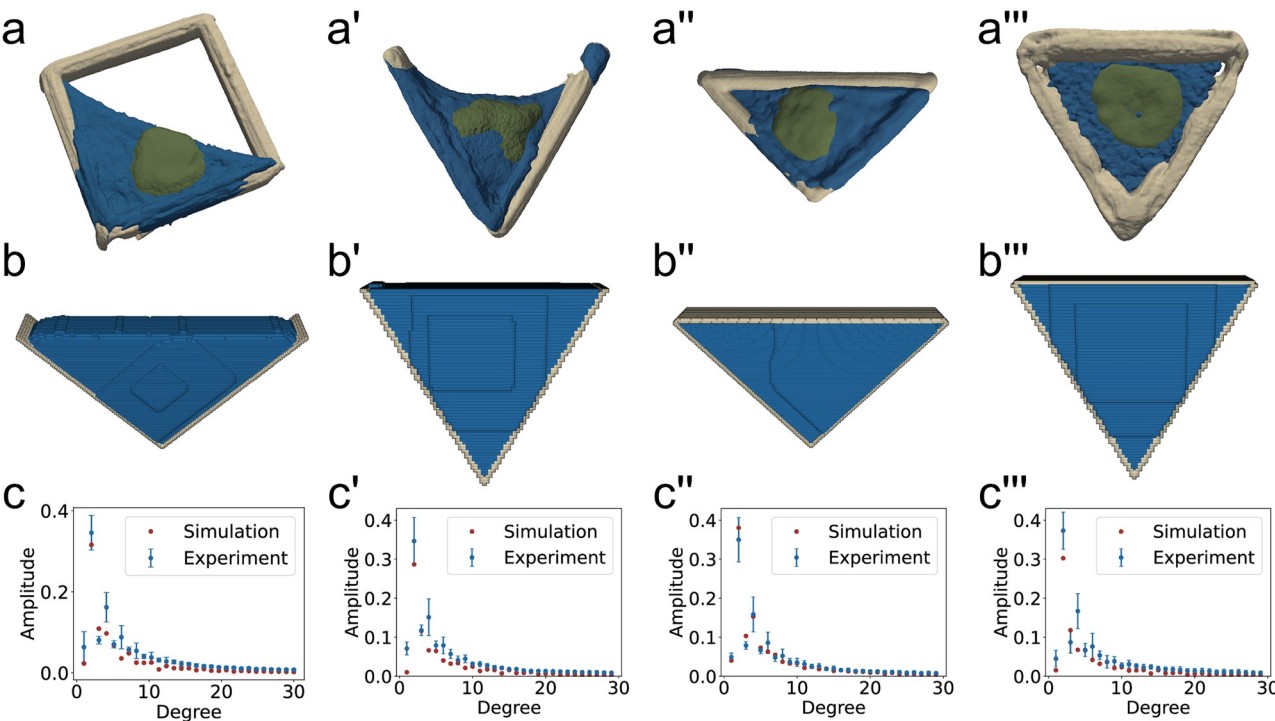

**Fig 7. Experimental and simulated shapes and frequency spectrum. a-a′′′**) Reconstruction of the experimental data from cells in L-shaped, V-shaped, right-angle triangular and equilateral triangular shaped structures. **b-b′′′**) Simulated cell shapes in structured environments. **c-c′′′**) Frequency spectra of the experimental and simulated cell shapes. **c)**$\Delta_{30} = 0.039$, **c′)**$\Delta_{30} = 0.136$, **c′′)**$\Delta_{30} = 0.053$ **c′′′)**$\Delta_{30} = 0.141$.

find that the predicted constant mean curvature (CMC) shapes agree well if the cell does not form invaginated arcs and stress fibers. Larger differences were found for cell shapes in L- and V-shaped structures, which are characterized by exposed areas and invaginated arcs.

In general, better cell shape prediction was possible with the CPM, independent of the choice of Hamiltonian and specific parameter values. Invaginated arcs form without explicitly simulating stress fibers and the high quadrupole moment in the spherical harmonics is correctly predicted. There are multiple factors that contribute to the better performance of CPM-simulations compared to surface minimization models. By selecting the neighbor order in the CPM, one can define how local the interaction energy is. This influences to which extend the surface minimization can occur, or may even facilitate surface extension between two general cell types, if the interaction energy is negative. A cell described by a Hamiltonian with volume, area, interaction and nucleus centering energies leads to more complex shapes that better resemble experimental results than the case in which only surface tension is minimized. In the future, it would be interesting to convert the CPM model parameters established here into physical values. As a first estimate, we note that the known cellular surface tension around 1 $nN/\mu m$ [17] should correspond to the interaction parameters J divided by pixel size squared. With our pixel size around 1 $\mu m$, the simulation temperature $T = 100$ thus corresponds to an energy scale of 100 $nN\mu m$, which makes sense because several focal adhesions are involved here, each with typical force 5 nN and typical size $\mu m$ [52]. For a more detailed analysis of the CPM-parameters, however, one had to conduct simulations with different resolutions [25].

Even though CPM-simulations better capture experimental cell shapes, it should be noted that there are some systematic differences between the simulated and experimental shapes. For

instance, cells in experiments have more variability in their shapes (compare Fig 2c) and attach only partially to the structures compared to the smooth shapes predicted here by the CPM when averaging over many configurations. In the future, one should investigate how well the single CPM-trajectories, which were not analysed here, can represent these fluctuations. To this end, it might be interesting to replace the Metropolis algorithm by a more physical algorithm, possibly even by one that represents non-equilibrium physics [53]. We also note the reconstruction of cell shape in experiments using actin staining is only one possible measure of cell shape, which in the future could be complemented e.g. by membrane stains. We finally note that actin stress fibers often span between adhesive sites, however their impact has not been accounted for in the simulations and could be included in the future, as done before for single cell CPM-simulations in 2D [18].

The best agreement between experimental and simulated cell shape was found with the linear area Hamiltonian, despite the fact that less parameters are used during the simulation. This indicates that cells in 3D scaffolds are not subject to strong homeostasis of their surface area, as it is known for cells adhering from solution onto flat substrates, when cell surface area can increase by large amounts. The effect of the nucleus on cell shape seems to be negligible in our setting and the nucleus is strongly deformed by the cytoskeleton, suggesting that there are mechanisms to keep it away from the cell surface and its mechanics, as represented here by the nucleus anchoring term in the CPM-Hamiltonian. This aspect could however change if the geometrical environment became even more restrictive. In the future, our results could be used for better understanding and even controlling cells in 3D scaffolds, which hold great promise to normalize cell shape and function.

## Supporting information

**S1 Appendix. Cell shapes simulated with the CPM without explicit representation of the nucleus.**
(PDF)

**S1 Table. Parameters for the CPM simulations.**
(PDF)

**S1 Video. Shape of an experimentally observed single cell in an L-shaped structure (left) and minimized surface in an L-shaped structure (right).**
(GIF)

**S2 Video. Shape of an experimentally observed single cell in a V-shaped structure (left) and minimized surface in a V-shaped structure (right).**
(GIF)

**S3 Video. Shape of an experimentally observed single cell in a right-angle triangular structure (left) and minimized surface in a right-angle triangular structure (right).**
(GIF)

**S4 Video. Shape of an experimentally observed single cell in an equilateral triangular structure (left) and minimized surface in an equilateral triangular structure (right).**
(GIF)

**S5 Video. Cellular Potts model simulation of a single cell in an L-shaped structure with the elastic area Hamiltonian (Eq 1).**
(MP4)

**S6 Video. Cellular Potts model simulation of a single cell in an L-shaped structure with the linear area Hamiltonian (Eq 2).**
(MP4)

**S7 Video. Result of the cellular Potts model simulation of a single cell in an L-shaped structure with the linear area Hamiltonian (Eq 2).**
(GIF)

## Author Contributions

**Conceptualization:** Rabea Link, Mona Jaggy, Martin Bastmeyer, Ulrich S. Schwarz.

**Data curation:** Rabea Link, Mona Jaggy.

**Formal analysis:** Rabea Link.

**Funding acquisition:** Martin Bastmeyer, Ulrich S. Schwarz.

**Investigation:** Rabea Link, Mona Jaggy.

**Methodology:** Rabea Link, Mona Jaggy.

**Project administration:** Martin Bastmeyer, Ulrich S. Schwarz.

**Resources:** Martin Bastmeyer, Ulrich S. Schwarz.

**Software:** Rabea Link.

**Supervision:** Martin Bastmeyer, Ulrich S. Schwarz.

**Validation:** Rabea Link, Mona Jaggy.

**Visualization:** Rabea Link, Mona Jaggy.

**Writing – original draft:** Rabea Link.

**Writing – review & editing:** Rabea Link, Mona Jaggy, Martin Bastmeyer, Ulrich S. Schwarz.

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
