## [Decision Letter · Decision Letter 0]

20 Nov 2023

Dear Prof. Dr. Schwarz,

Thank you very much for submitting your manuscript "Modelling Cell Shape in 3D Structured Environments: A Quantitative Comparison with Experiments" for consideration at PLOS Computational Biology.

As with all papers reviewed by the journal, your manuscript was reviewed by members of the editorial board and by several independent reviewers. In light of the reviews (below this email), we would like to invite the resubmission of a significantly-revised version that takes into account the reviewers' comments.

You will find two extensive reviewer reports on your manuscript. Both these reviewers are experts in the topic of the paper and I hope you will find these comments useful in revising your paper. While reviewing your manuscript, I would also request that you pay special attention to the narrative so that the different model assumptions and comparisons are made clear. We look forward to receiving your revision.

We cannot make any decision about publication until we have seen the revised manuscript and your response to the reviewers' comments. Your revised manuscript is also likely to be sent to reviewers for further evaluation.

Sincerely,

Padmini Rangamani

Academic Editor

PLOS Computational Biology

Daniel Beard

Section Editor

PLOS Computational Biology

Dear Authors,

You will find two extensive reviewer reports on your manuscript. Both these reviewers are experts in the topic of the paper and I hope you will find these comments useful in revising your paper. While reviewing your manuscript, I would also request that you pay special attention to the narrative so that the different model assumptions and comparisons are made clear. We look forward to receiving your revision.

Reviewer's Responses to Questions

**Comments to the Authors:**

Reviewer #1: In the manuscript entited "Modelling cell shape in 3D structured environments: a quantitative comparison with experiments" by Link et al, the authors use variations of two modeling strategies to reproduce geometries of cells grown in 3D laser printed scaffolds of various shapes. I note to the editor that I cannot comment on the validity or approach of the the experimental methods. The two modeling approaches are to perform surface relaxation to approach a constant mean curvature surface (CMC), and a cellular Potts model (CPM). The authors find that the outputs from surface relaxation using Surface Evolver deviate from the observed geometries. Performing parameter variations for the CPM produces geometries with improved correspondence with the experimental geometries. The modeling efforts appear to be technically sound, although additional care to modeling details and comparisons is warranted. Further discussion on the biological appropriateness of the parameters they identify could improve the strength of conclusions of this work. The work is overall of interest to the readership of PLOS Computational Biology.

Major points:

1. (SI Table 1) What are the units for values in this table?

2. How do the best fit parameters for the CPM model compare against experimental properties? Is there anything to be learned from the strength of adhesions, surface tension, etc.?

3. (Line 176) Since the simulation is stochastic monte carlo, the system will tend towards lower energy states but there will be fluctuations around a local minima (c.f. supplemental movies). Moreover, depending on the energy landscape and the magnitude of fluctuations, a system can be kinetically trapped and thus prevented from reaching the "lowest energy state". The statement should be amended accordingly.

4. Related to major point 3, how was the representative output of each CPM simulation selected? The minimal energy configuration observed from the trajectory?

5. (Section Comparison to CPM-Shapes) The initial configuration of the cellular Potts model is not clear. From the supplemental movies, it appears that an idealized scaffolding structure is used. For quantitative comparison using Δ30 against the CMC results, it would be beneficial to use similar inputs and constraints. Alternatively differences between workflows should be stated and possible limtations discussed.

6. What are the reported differences between CPM and experimental geometries. Are they also mean Δ30 values as reported for CMC results?

7. Uncertainty quantification of Δ30 should be shown where possible for modeled geometries against experiment. If possible, statistical hypothesis tests could also be used to guide interpretation of Δ30 differences.

8. Establishing a baseline or threshold for Δ30 values may also be helpful for interpreting the modeling results. For example, an NxN heatmap of Δ30 for experimental geometries of cells grown in a common scaffold against each other could give a sense of variability in shape from cell to cell. Further it would be beneficial to show results from several cells or models.

9. (Data availability) Imaris, a commercial software, is used to extract triangulated meshes. It would benefit reproducibility to share the output triangulations, input micrographs, and segmentations.

Minor points:

1. (Line 325) Table S1 is missing "S1"

2. (Discussion) The authors suggest that their approach could be used for rational scaffold design despite that they found limited transferrability of parameters from scaffold to scaffold. I think some discussion of biological correspondence of their parameters and/or what may be missing from the model to lead to universal parameters could benefit their claim.

Reviewer #2: Summary:

This paper describes a systematic comparison of cell shapes grown in 3D structured environments with different modeling approaches – an energy-minimizing Constant Mean Curvature (CMC) surface model, and different variants of a cellular potts model (CPM). They analyze the 3D shapes with a 3D spherical harmonics analysis to get the frequency spectra that can be directly compared between experiment and model. Comparison with experiment demonstrates that the CPM area energy approach captures the shape better. There is some discussion of the 3D scaffold production, and the rational design of scaffolding in the future, but this reviewer thinks it muddles the more obvious narrative of the paper. While the narrative can be improved, the content of the paper is exciting and worthy of publication after minor revisions.

Specific comments:

Overall:

- The description of the 3D scaffold production with laser nano-printing seems over-emphasized in the abstract/introduction and should be framed more as supporting information for the shape analysis. The main concern of the 3D scaffolds should be why these particular four shapes were chosen and how they influence the resulting cell shape analysis.

- The analysis, which exploits the particular scaffold shape for simplifying the coordinate system, should also be better justified. Are these regular-shaped scaffolds over-simplifying the shape of cells which would likely be much more complex in natural environments?

- Since the main purpose of this paper is to compare experiments with models, it should focus more explicitly from the start on comparing the different model methods. It would make for a stronger narrative if there was a more systematic introduction to the modeling approaches, their assumptions, and their algorithms.

Abstract:

- “3D scaffolds for cell adhesion were manufactured using 3d laser nano-printing and the shapes of single mesenchymal NIH/3T3 cells were compared by a Fourier method with surfaces that minimize area …” This makes it sound like the 3D scaffolds are compared with the shapes of the cells. Please reword more carefully, separating out the different important concepts.

- “Rational design of scaffolds for desired cell shape” Is this paper more about designing the scaffolds, or making conclusions about cell shapes by comparison with modeling? The way this is introduced in the abstract will influence the reader’s evaluation of this paper.

3D structured environments:

- Add labels for the four shapes in Fig 1a to make these names less ambiguous when reading the text. “L-shaped” and “V-shaped” with “right triangle” or “equilateral triangle” are currently used, please reuse these same names throughout the text.

Constant Mean Curvature (CMC) surfaces:

- This needs to be more clearly introduced as one of the modeling approaches when described in methods. As it is currently written, it can easily be interpreted as a step of the analysis.

Cellular Potts Model (CPM):

- I would like to see a more explicit treatment of the different variants of CPM used. Currently these are addressed in one long paragraph. This needs at least one paragraph for the elastic energy assumptions and model, and one for the linear area energy assumption and model. It would be useful to assign these explicit names, so they can be referenced later in the paper. Also mention how many simulations were run, what were the parameters, and any other important metadata if anyone wants to reproduce these simulations.

- “Cellular Potts models (CPMs) are a more evolved approach to model single cells” What do you mean by “more evolved”?

3D Spherical Harmonics Analysis:

- These particular scaffold shapes, combined with the spherical harmonics analysis, allow the authors to exploit the star-shapes to map to a simplified unit sphere. This seems like it might be oversimplifying the analysis of cell shape to a very restricted domain.

- Please describe why you chose to use the first 30 degrees of the respective frequency spectra. What kinds of shape features do you expect to detect in this range?

Results:

- Fig 2. Call this something more obvious like “Comparison between experimental cell shapes and minimized surface model”

**Have the authors made all data and (if applicable) computational code underlying the findings in their manuscript fully available?**

Reviewer #1: **No: **The authors note that scripts and data will be made available on Zenodo upon publication. It is not clear if the input experimentally derived cell shapes nor raw output data are included in the deposition.

Reviewer #2: Yes

PLOS authors have the option to publish the peer review history of their article (what does this mean?). If published, this will include your full peer review and any attached files.

Reviewer #1: No

Reviewer #2: No
---

## [Decision Letter · Decision Letter 1]

14 Mar 2024

Dear Prof. Dr. Schwarz,

We are pleased to inform you that your manuscript 'Modelling Cell Shape in 3D Structured Environments: A Quantitative Comparison with Experiments' has been provisionally accepted for publication in PLOS Computational Biology.

Best regards,

Padmini Rangamani

Academic Editor

PLOS Computational Biology

Daniel Beard

Section Editor

PLOS Computational Biology

Reviewer's Responses to Questions

**Comments to the Authors:**

Reviewer #1: For the revision of the manuscript entitled "Modelling cell shape in 3D structured environments: A quantitative comparison to experiments", Link and colleagues have suitably addressed the comments from the previous review.

I suggest to the authors that a minor proofreading could be beneficial but have otherwise no further concerns.

Reviewer #2: Summary:

The authors have made substantial, meaningful improvements to their paper, highlighted by the addition of a new figure, enhanced comparisons, a more robust presentation of model details and findings, and discussion of the physical effects underlying cell shape. This systematic comparison of different modeling methods with each other and with data on cell shape is novel, rigorous, and sets a standard for future studies. The work is of interest to PLoS CB and I recommend its publication with just one very minor improvement detailed below. Congratulations on writing a very nice paper.

Specific comments:

- The CPM formulation (equation 1) includes a third term with interaction energies. Since this paper only simulates single cells there should be no interactions with neighbor cells, leading me initially to think this term should always be 0. After inspection of Table S1 I see there are interactions between cell and medium, medium and nucleus, etc. The authors should mention this when introducing the interaction energies in the CPM model.

- Table S1 refers to Jcell,medium. I believe this is referred to as Jmedium,cytoplasm throughout the text (especially in Comparison to CPM-shapes). Please choose one and keep it consistent.

**Have the authors made all data and (if applicable) computational code underlying the findings in their manuscript fully available?**

Reviewer #1: Yes

Reviewer #2: Yes

PLOS authors have the option to publish the peer review history of their article (what does this mean?). If published, this will include your full peer review and any attached files.

Reviewer #1: No

Reviewer #2: No

---

## [Editor Report · Acceptance letter]

28 Mar 2024

PCOMPBIOL-D-23-01254R1 

Modelling Cell Shape in 3D Structured Environments: A Quantitative Comparison with Experiments

Dear Dr Schwarz,

I am pleased to inform you that your manuscript has been formally accepted for publication in PLOS Computational Biology. Your manuscript is now with our production department and you will be notified of the publication date in due course.

With kind regards,

Anita Estes
